# Analysis of Calprotectin as an Early Marker of Infections Is Economically Advantageous in Intensive Care-Treated Patients

**DOI:** 10.3390/biomedicines11082156

**Published:** 2023-08-01

**Authors:** Aleksandra Havelka, Anders O. Larsson, Johan Mårtensson, Max Bell, Michael Hultström, Miklós Lipcsey, Mats Eriksson

**Affiliations:** 1Department of Molecular Medicine and Surgery, Karolinska Institute, 171 76 Stockholm, Sweden; aleksandra.havelka@gentian.no; 2Gentian Diagnostics AS, 1596 Moss, Norway; 3Department of Medical Sciences, Clinical Chemistry, Uppsala University, 751 85 Uppsala, Sweden; 4Department of Physiology and Pharmacology, Section of Anaesthesiology and Intensive Care Medicine, Karolinska Institute, 171 77 Stockholm, Sweden; 5Department of Perioperative Medicine and Intensive Care, Karolinska University Hospital, 171 76 Stockholm, Sweden; 6Department of Surgical Sciences, Section of Anaesthesiology and Intensive Care Medicine, Uppsala University, 751 85 Uppsala, Sweden; 7Department of Medical Cell Biology, Integrative Physiology, Uppsala University, 751 23 Uppsala, Sweden; 8Department of Epidemiology, McGill University, Montréal, QC H3A 0G4, Canada; 9Lady Davis Institute of Medical Research, Jewish General Hospital, Montréal, QC H3T 1E2, Canada; 10Hedenstierna Laboratory, Department of Surgical Sciences, Uppsala University, 751 85 Uppsala, Sweden; 11NOVA Medical School, New University of Lisbon, 1099-085 Lisbon, Portugal

**Keywords:** calprotectin, costs, early detection, economic modeling, infection, intensive care, sepsis

## Abstract

Calprotectin is released from neutrophil granulocytes upon activation. Several studies have indicated that plasma calprotectin is an early determinant of bacterial infections, which may serve as a diagnostic tool facilitating decision making on antibiotic treatment. The study objective was to explore the health and economic implications of calprotectin as a predictive tool to initiate antimicrobial therapy in a cohort of critically ill patients. Thus, data obtained from a previously published study on calprotectin as a hypothetical early biomarker of bacterial infections in critically ill patients were evaluated regarding the potential cost-effective impact of early analysis of calprotectin on an earlier start of antibiotic treatment. Under the assumption that calprotectin is used predictively and comparators (white blood cells, procalcitonin, and C-reactive protein) are used diagnostically, a cost-effective impact of EUR 11,000–12,000 per patient would be obtained. If calprotectin would be used predictively and comparators would be used predictively for 50% of patients, it is hypothesized that cost-effectiveness would be between EUR 6000 and 7000 per patient, based on reduced stay in the ICU and general ward, respectively. Furthermore, predictive use of calprotectin seems to reduce both mortality and the length of hospital stay. This health economic analysis on the predictive use of plasma calprotectin, which facilitates clinical decision making in cases of suspected sepsis, indicates that such determination has a cost-saving and life-saving impact on the healthcare system.

## 1. Introduction

Globally, the estimated number of sepsis cases in 2017 was nearly 50 million, and there were approximately 11 million sepsis-related deaths [1]. Deaths in severe sepsis and septic shock are not restricted to the acute phase but remain high even three months after diagnosis [2]. One of the most important aspects of the management of patients with sepsis is early recognition so that the administration of antibiotics, source control measures, and effective resuscitation strategies can be initiated as soon as possible [3].

Prompt administration of antibiotics is associated with improved outcomes in sepsis and septic shock. Every 1 h delay in antibiotics after emergency department (ED) triage or the onset of organ dysfunction or shock may lead to a 3–7% increase in the odds of a poor outcome [4,5,6]. A quality improvement program has resulted in a significant reduction in in-hospital sepsis cases and sepsis mortality and yielded a considerable positive return on investment [7], showing that improvements can be achieved. The mean total hospital costs per patient with sepsis vary considerably between countries. The relative amount of healthcare budget spent on sepsis per country is estimated at 2.65%, which equals a median (interquartile range (IQR)) amount of gross national product spent on sepsis-related healthcare of 0.33% (0.03–2.27%) [8]. Early identification of patients with sepsis is a prerequisite to triggering healthcare interventions to improve outcomes. Cost-effective biomarkers with rapidly available results around the clock and high specificity and sensitivity could improve the management of septic patients.

In severe infections, neutrophil granulocytes adhere to and penetrate vascular walls, which reduces their number in the blood and reduces the predictive value of the number of circulating neutrophil granulocytes [9]. Calprotectin, a heterodimer of two protein subunits, is present in high concentrations in the cytoplasm of neutrophils and is released into the circulation upon neutrophil activation without the requirement for de novo synthesis.

Several studies have confirmed the fast release of calprotectin upon inflammatory stimuli and earlier detection of calprotectin in circulation compared to procalcitonin (PCT) and C-reactive protein (CRP) [10,11].

Calprotectin release weakens cellular adhesion, favoring leukocyte extravasation, and activates pro-inflammatory pathways regulated by the cell surface receptors TLR 4 and RAGE, leading to the expression of pro-inflammatory cytokines and amplification of the inflammatory response. CRP expression is induced by interleukin-6 (IL-6) and occurs in the liver, which results in slower kinetics and late detection of this biomarker in the circulation. IL-6 is rapidly but transiently expressed in response to infection, inflammation, and tissue injury [12]. Calprotectin has been shown to be a valuable biomarker for early detection of infections [9,10,11], and its soluble form provides both bacteriostatic and cytokine-like effects in the local environment [13].

Calprotectin, analyzed at admission to the intensive care unit (ICU), has a high sensitivity for bacterial infections in severely ill patients and has predictive value for mortality [14,15,16]. Previous studies confirm the value of calprotectin in the identification of bacterial infections, with better sensitivity and specificity than CRP and PCT [14,15,17,18]. Moreover, calprotectin was superior to PCT when patients with and without bacterial sepsis were distinguished [15].

Based on the results from previous studies, we hypothesized that identifying patients with severe infections/sepsis at an early stage using calprotectin would be beneficial for patients and healthcare providers and economically superior to other commonly used biomarkers.

Consequently, the primary aim of the present study was to evaluate whether analysis of calprotectin might be an economically beneficial complement to conventionally used biomarkers of infection, i.e., CRP, white blood cell count (WBC), and PCT, in a previously published cohort of unselected ICU patients [16], where the aim was to assess plasma calprotectin as an early marker of bacterial infections in critically ill patients in comparison with PCT, CRP, and WBC.

## 2. Materials and Methods

### 2.1. Clinical Setting and Target Population

This health economic analysis is based on a study published by Jonsson et al. [16], aiming to assess the value of plasma calprotectin as an early biomarker of bacterial infections in critically ill patients. One hundred and eighty-eight eligible patients with an expected length of stay (LOS) > 24 h admitted to the General Intensive Care Unit (ICU) at Karolinska University Hospital, Solna, Sweden, were evaluated for inclusion in the study. A flowchart showing the patient selection path has previously been published [16], showing 110 included patients, out of which 58 patients (52.7%) had antibiotic therapy initiated during their ICU stay. To avoid potential misclassification in our analysis, we excluded 20 patients with a possible or probable infection. Therefore, we included 58 patients with no infection and 38 patients with a confirmed infection in the final analysis. Initial blood samples were collected on ICU admission. The results of the calprotectin analyses were blinded to an infectious disease specialist who determined the likelihood of infection.

In the present hypothetical study on the cost-effectiveness of calprotectin, calculations were based on the findings of Jonsson et al. [16]. Their study was approved by The Regional Ethics Review Board in Stockholm, Sweden (2006/1469-31/2 and subsequent amendments) and performed in accordance with ethical principles that have their origin in the Declaration of Helsinki [19], which are consistent with ICH/Good Clinical Practice (GCP) E6 (R2), and EU Clinical Trials Directive, and applicable local regulatory requirements. Since the present study consists of de-identified data in which patients could not be individually identified during the analysis process; patient consent was not required. Our results are based on a theoretical model using an algorithm for mathematical evaluation, without any new collection of clinical data. Hence, renewed ethical permission was not necessary [20].

### 2.2. Analyses of Biomarkers

At ICU admission, or shortly thereafter, and on consecutive days, up to one week, blood samples were collected in EDTA tubes and centrifuged at 2000 rpm for 10 min at +4 °C. Supernatants were stored at −80 °C until analyzed at the Department of Clinical Chemistry, Uppsala University Hospital, Uppsala, Sweden. Calprotectin was analyzed using a Mindray BS-380 (Mindray Medical International, Shenzhen, China) and a particle-enhanced turbidimetric immunoassay, (GCAL^®^ Calprotectin Immunoassay) from Gentian AS (Moss, Norway). CRP was analyzed on an Architect Ci8200 (Abbott Laboratories, Abbott Park, IL, USA). The expected normal CRP level was <5 mg/L. PCT was analyzed using Cobas EE (Roche Diagnostics, Mannheim, Germany). The expected normal PCT level was <0.05 ng/mL.

### 2.3. Base Data and Model Structure

A decision tree model is intended to capture the initial outcomes of two or more decisions without considering time as a component of the outcome of the decisions. Since the intervention (calprotectin analysis) and the comparators have the same intended outcomes within a limited period of time, a decision tree model was chosen as the most appropriate for calculating the cost-effectiveness of calprotectin.

The decision tree shows the different diagnostic outcomes of correctly or incorrectly diagnosing a bacterial infection. The same cohort of patients is subjected to either detecting or not detecting the infection, which determines the timing of the antibiotic treatment and the outcomes (survival, length of hospital stay, and costs). The sensitivity of the tests determines the proportion of patients who are correctly (true positive) and incorrectly (false positive) tested for having a bacterial infection, and the specificity determines those who are correctly and incorrectly diagnosed as not having a bacterial infection (true negative and false negative, respectively). The structure of the decision tree is presented in Figure 1.

The patient pathway in the decision tree for a predictive test starts with defining the population (patients admitted to the ICU); however, this does not in itself impact the pathway. The sensitivity and specificity of the predictive biomarker test will determine what proportion of the patient population is correctly or incorrectly diagnosed. The proportion of those who are correctly diagnosed will be predictively detected with a bacterial infection and receive antibiotic treatment 24 h prior to clinical presentation and the subsequent antibiotic treatment. The proportion correctly diagnosed as not having a bacterial infection (true negative) will not be treated with antibiotics. The proportion of the patient population that is incorrectly diagnosed will not be detected before clinical presentation and, thus, will receive antibiotic treatment after clinical presentation. The patient population that is incorrectly diagnosed where the predictive test suggests bacterial infection (false positive) will receive antibiotic treatment together with the patients who are predictively detected correctly. Based on the pathway, patients face varying outcomes (mortality, length of stay), resulting in different costs and survival rates.

Figure 2 illustrates the decision tree structure when there is no test taken. In this case, the proportion of patients with an infection will not be detected before clinical presentation, which will result in a longer length of stay and an increased risk of mortality.

This model will estimate the cost-effectiveness of calprotectin analysis for early detection of bacterial infections as compared to analysis of WBC, procalcitonin, CRP, and/or conducting no test at all in a Swedish intensive care setting. Outcomes include costs, mortality rates, and length of stay at hospital. The timing of the first administration of antibiotics affects the total length of stay in the hospital [6]. In this scenario, it is assumed that early detection of sepsis implies a total hospital stay of 15 days, while symptomatic detection is associated with 30 days of hospitalization.

The comparators included in the analysis are WBC, PCT, CRP, and no testing. The analysis is based on patients admitted to an ICU in Sweden, with inclusion and exclusion criteria from the literature applied [16].

Table 1 presents the predictive accuracy of detecting a bacterial infection by the respective biomarker 24 h before initiating antibiotic treatment. The diagnostic accuracy of the biomarker test is presented in Table 2. The diagnostic test is taken on the same day as the initiation of the antibiotic treatment when the patient is symptomatic.

As evaluated by the areas under the receiver-operating characteristics (ROC) curves, calprotectin predicted bacterial infections at 0.82 (95% CI, 0.70–0.94), whereas all the other biomarkers included in the study had lower ROC areas [16].

### 2.4. Diagnostic Yield

The diagnostic yield is the probability of a patient having a positive diagnosis, i.e., a patient is diagnosed to have a bacterial infection and needs antibiotic treatment. This does not only depend on the sensitivity and specificity outlined above but also on the number of sequential tests performed, whether the test is taken predictively or diagnostically, and if the physician decides to initiate antibiotic treatment or not after analyzing the test results.

The model allows for a total of 10 sequential tests, distributed as either predictive or diagnostic. Since the predictive performance of GCAL^®^ is central to the CE analysis, 100% of patients in the GCAL arm will undergo predictive tests and 0% in the comparator arms (which can be changed between 0 and 100% by the user). The test sensitivity and specificity are adjusted accordingly to the proportion of patients who are assumed to use the test predictively, i.e., in the base case, no patients are identified predictively for the comparators. The estimated treatment cost per day in the ICU is EUR 5530 and EUR 591 in a general ward (see “Limitations” section).

The model will by default assume that the physician will trust the test if the test result suggests a bacterial infection and initiate the antibiotic treatment early. In cases where tests are used predictively and the test result is negative, another test will be taken when the patient becomes symptomatic of infection. The model assumes by default that the physician will perform a diagnostic test and initiate antibiotic treatment if the patient is symptomatic. This confirmatory test affects the overall sensitivity and specificity.

The base-case settings and the diagnostic yield result (final sensitivity and specificity) for the base case are presented in Table 3. Notice the comparators remain equal to Table 2 as they are used diagnostically; however, the calprotectin values are updated due to the possibility of receiving a false negative result (as previously described).

The length of the ICU stay was estimated by an algorithm defined by Ferrer et al. [6] and Paoli et al. [21], in which they stratified the unadjusted length of stay (LoS) in an ICU by the timing of the antibiotic therapy. In patients presenting with sepsis in the emergency department, the linear relationship is as follows: LoS = 8.4143 + 0.7714 × hours to antibiotic therapy, whereas mortality (%) was equal to 0.2291 + 0.0241 × hours to antibiotic therapy [21]. However, the Ferrer et al. study [6] only includes data on antibiotic administration from 0 to 6+ h after admission to an ICU, and no hour-specific data were presented after 6 h. It is also important to note that the Ferrer et al. data [6] suggest that patients who received treatment in the first hour are most likely to have more severe symptoms present. Therefore, the LoS algorithm has a lower bound of 1 h and an upper bound of 6 h for antibiotic treatment. As the Ferrer et al. study [6] was not performed in a Swedish setting, an adjustment of the equation estimating LoS was considered necessary. Thus, we applied a multiplier of 0.5 in order to utilize a safety margin.

Symptomatic detection is assumed to be equal to the upper bound of the Ferrer algorithm [6], and thus, early detection is assumed to be 6 h before symptomatic detection. The LoS for early detection of infection and symptomatic detection following the algorithm and applied adjustment are presented in Table 4.

## 3. Results

In the base-case scenario, calprotectin is employed predictively, differently from the comparators that are employed diagnostically. Under these assumptions, it is suggested that the analysis of calprotectin in patients with suspected sepsis saves costs and reduces in-hospital mortality in those patients. The mean duration of in-patient care is consequently reduced compared to all comparators (Table 5a,b).

Currently, no alternative settings for various inputs have been identified besides the severity adjustments available in the model. Thus, arbitrary ranges of assumed key inputs have been used to assess the impact these inputs have on the model results.

When including a predictive test for 50% of patients in the comparator scenarios (except the no-test option), analysis of calprotectin remains the dominant option (Table 6a,b).

The calprotectin assay has been shown to detect infection up to 24 h prior to symptomatic onset [16]. However, the linear relationship in the Ferrer et al. algorithm [6] cannot be extended beyond the 6 h difference, as doing so equates to a negative LoS for 24 h early detection. Therefore, to assess the impact of earlier mean time to antibiotics by determining plasma calprotectin, the LoS at an ICU for early detection is varied from 0 days to a maximum of 6.5 days, which is equal to the LoS for symptomatic detection. Table 7 displays the results of these analyses. No changes in the comparators are observed, as they are not used predictively in these cases. Thus, they were not included in the table.

Estimated differences in LoS in a general ward (G. ward) for early and symptomatic detection are shown in Table 8 and Table 9, respectively. In these assumptions, the total length of in-patient care for early and symptomatic patients is 15 days and 30 days, respectively. These total LoS equate to 10.8 days and 23.5 days spent in a general ward for early and symptomatic treated patients. As these values are inherently uncertain, sensitivity analysis surrounding the total LoS and the related LoS in a general ward is assessed. Note that the LoS in an ICU remains the same as originally assumed. Analysis of calprotectin remains the principal option.

## 4. Discussion

This analysis, which was performed by an independent company [22], specializing in health economic research, is essentially based on a study indicating that plasma calprotectin is useful as an early complementary biomarker of bacterial infections in ICU patients [16], where WBC, PCT, and CRP may still provide valuable information. Hence, the advantage of calprotectin in this context assumes a fast increase in this biomarker in the circulation and, thereby, earlier detection of infection when compared to other routinely used biomarkers. Several other studies have also concluded that early detection of bacterial infections facilitates decision making on antimicrobial therapeutic interventions, which may improve outcomes [4,5,11,14,15,17,18,23,24]. Although this study focuses on a health economic perspective, the main rationale for the analysis of calprotectin is from a clinical perspective, since early diagnosis of severe infections and sepsis reduces both delays in treatment and mortality [5,24,25,26]. From this aspect, our findings are in alignment with previous ones, indicating that early detection of severe infections and sepsis has both a cost-saving and a life-saving impact in the ICU setting [7,21].

An estimated median (IQR) total hospital cost per patient stay for sepsis is EUR 32,421 (USD 20,090), depending on the country and type of hospital. The median reported hospital-wide costs (IQR) of survivors and non-survivors were EUR 34,855 (USD 14,352) and EUR 20,537 (USD 17,817), respectively [25] (USD/EUR ~0.95). In a recent study on cost-effectiveness in the ICU, it was deduced that if sepsis could be detected 3 h earlier than current practice, this would result in substantial annual cost savings for the healthcare system [26].

In a U.S. cohort of patients developing sepsis during hospital stay but not diagnosed on admission, the mean overall cost was more than EUR 51,022, whereas the mean cost for those diagnosed with sepsis on admission was EUR 18,023 [23]. When sepsis was not diagnosed on admission, the mortality rate was twice that seen when sepsis was diagnosed on admission. However, it cannot be excluded that higher costs and a poorer prognosis, at least to some extent, may be attributable to some patients developing sepsis during the hospital stay.

Calprotectin has also been shown to be a promising biomarker for estimation of disease severity, prediction of organ failure, and mortality in patients with severe infections, including bacterial infections and SARS-CoV-2 infections presented at the emergency department as well as in an ICU setting [24,27,28,29]. Unlike other routinely used inflammatory biomarkers, such as CRP and PCT, calprotectin is released by neutrophils into the bloodstream without requiring de novo protein biosynthesis [27]. Calprotectin can today be measured on high-throughput instruments at the same time to result in CRP, which is important for acute diagnoses and optimal treatment decisions.

### Strengths and Limitations

Although this is a retrospective study, the original data [16] were collected prospectively. Calprotectin was sampled daily and was significantly elevated in infected patients. Since therapeutic interventions were not steered by calprotectin levels, this indicates that the prediction of calprotectin as a biomarker of bacterial infection was unbiased. We excluded patients with uncertain infection status, i.e., patients with a possible or probable infection, to avoid misclassification. However, such patients are common in the ICU and may also benefit from early antibiotic therapy.

The most obvious limitation of this study is that the results are due to the use of an algorithm. However, the used algorithm is based on international data sources [6,21,23] and with slight modifications to fit with Swedish clinical practice, which provides a more accurate estimate than simpler models. The assumption that calprotectin should have been taken predictively, whereas the comparators were analyzed based on clinical indications, is also a limitation. A better approach would have been a time-dependent analysis, which, however, is not possible retrospectively. Estimations of hospital costs were performed in a health economic analysis reported by Quantify Research AB, Stockholm, Sweden [22] and based on publicly available information ((in Swedish); SEK 1/EUR ~0.09). Since both hospital costs and exchange rates vary over time, calculated expenses are rounded off to the closest thousandfold number just below the estimated price.

It is a drawback that the diagnostic accuracy of calprotectin, measured in combination with conventionally analyzed biomarkers (i.e., WBC, PCT, and CRP), was not calculated. However, it is tempting to speculate that algorithms managed by artificial intelligence may be useful for such calculations in the not-too-distant future.

## 5. Conclusions

In the present base-case scenario, calprotectin was identified as a cost-effective biomarker for an unselected patient cohort presenting in a Swedish ICU. Compared to procalcitonin, leucocytes, and CRP, it is suggested that calprotectin saves total costs, reduces the mean duration of in-patient care, and reduces in-hospital mortality in those patients. In the sensitivity analysis, calprotectin remains the dominant option when key model inputs are varied.

## Figures and Tables

**Figure 1 biomedicines-11-02156-f001:**
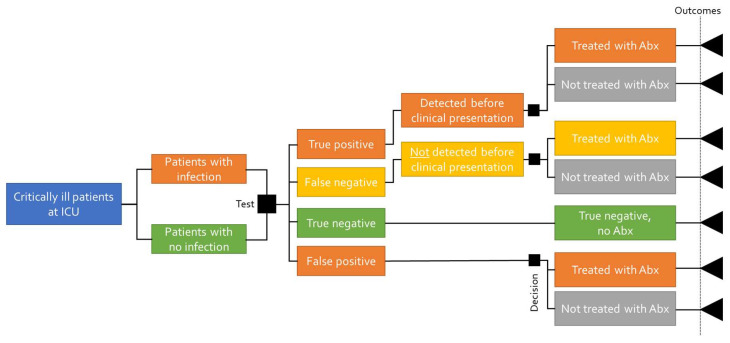
Decision tree structure with a predictive test.

**Figure 2 biomedicines-11-02156-f002:**
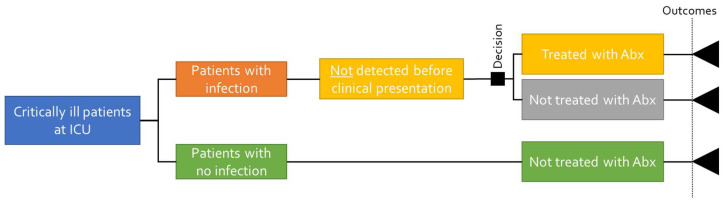
Decision tree structure with no diagnostic test.

**Table 1 biomedicines-11-02156-t001:** Sensitivity and specificity of predictive test (24 h prior to antibiotic prescription).

Biomarker (Test)	Sensitivity	Specificity	Source
Calprotectin immunoassay	66%	93%	[16]
WBC count	67%	41%	
Procalcitonin	58%	58%	
C-reactive protein (CRP)	58%	94%	
No test	0%	0%	Assumption

**Table 2 biomedicines-11-02156-t002:** Sensitivity and specificity of diagnostic test (same day as antibiotic prescription).

Biomarker (Test)	Sensitivity	Specificity	Source
Calprotectin immunoassay	55.0%	90.0%	[16]
White blood cell count	42.0%	74.0%	
Procalcitonin	73.0%	58.0%	
C-reactive protein (CRP)	82.0%	54.0%	
No test	0.0%	0.0%	Assumption

**Table 3 biomedicines-11-02156-t003:** Applied sensitivity and specificity in the base-case scenario.

Biomarker (Test)	Sensitivity	Specificity	Source
Calprotectin immunoassay	85%	99%	Calculations
WBC count	42%	74%	
Procalcitonin	73%	58%	
C-reactive protein (CRP)	82%	54%	
No test	0.0%	0.0%	Assumption

**Table 4 biomedicines-11-02156-t004:** Length of stay at ICU with the mathematical linear relationship.

	Default Days	Source
Early detection (6 h earlier)	4.2	[6,21]
Symptomatic detection	6.5	
	Share	Source
Adjustment of LoS	0.50	Assumed safety margin

**Table 5 biomedicines-11-02156-t005:** (a) Base-case outcomes. (b) Base-case outcomes (difference compared to calprotectin) per comparator.

(**a**)				
	**Total Cost [EUR (Rounded off)]**	**Deaths**	**Mean Days ICU**	**Mean Days General Ward**
Calprotectin immunoassay	25,000	0.2	3.7	8.0
WBC count	36,000	0.3	5.0	15.1
Procalcitonin	37,000	0.3	5.1	15.3
C-reactive protein (CRP)	37,000	0.3	5.2	14.9
No test	40,000	0.3	5.7	15.8
(**b**)				
	**Δ Total Cost [EUR (Rounded off)]**	**Δ Deaths**	**Δ Mean days ICU**	**Δ Mean Days Ward**
Calprotectin immunoassay	Reference	Reference	Reference	Reference
WBC count	11,000	0.1	+1.3	+7.1
Procalcitonin	12,000	0.1	+1.4	+7.3
C-reactive protein (CRP)	12,000	0.1	+1.5	+6.9
No test	15,000	0.1	+2.0	+7.8

**Table 6 biomedicines-11-02156-t006:** (a) Outcomes when using comparators predictively for 50% of patients instead of 0%. (b) Outcomes (difference compared to calprotectin) per comparator when using comparators predictively for 50% of patients instead of 0%.

(**a**)				
	**Total Cost [EUR (Rounded off)]**	**Deaths**	**Mean Days ICU**	**Mean Days General Ward**
Calprotectin immunoassay	25,000	0.2	3.7	8.0
WBC count	32,000	0.3	4.5	13.0
Procalcitonin	32,000	0.3	4.5	12.5
C-reactive protein	31,000	0.3	4.4	11.9
No test	40,000	0.3	5.7	15.8
(**b**)				
	**Δ Total Cost [EUR (Rounded off)]**	**Δ Deaths**	**Δ Mean Days ICU**	**Δ Mean Days Ward**
Calprotectin immunoassay	Reference	Reference	Reference	Reference
WBC count	7000	+0.1	+0.8	+5.0
Procalcitonin	7000	+0.1	+0.8	+4.5
C-reactive protein	6000	−0.1	+0.7	+3.9
No test	15,000	+0.1	+1.9	+7.8

**Table 7 biomedicines-11-02156-t007:** Variation in LoS at an ICU for early detection using GCAL^®^.

Early Detection LoS in an ICU	Total Cost [EUR (Rounded off)]	Deaths	Mean Days ICU	Mean Days General Ward
0 days	15,000	0.2	1.8	9.9
1 day	17,000	0.2	2.3	9.4
2 days	20,000	0.2	2.7	9.0
3 days	22,000	0.2	3.2	8.5
4 days	24,000	0.2	3.6	8.1
5 days	26,000	0.2	4.0	7.7
6 days	28,000	0.2	4.5	7.2
6.5 days [equal to symptomatic (Table 4)]	29,000	0.2	4.7	7.0

**Table 8 biomedicines-11-02156-t008:** Total LoS of 15 days for early and symptomatically treated patients.

	Total Cost [EUR (Rounded off)]	Deaths	Mean Days ICU	Mean Days General Ward
Calprotectin immunoassay	24,000	0.2	3.7	6.8
WBC count	31,000	0.3	4.9	7.2
Procalcitonin	32,000	0.3	5.1	7.4
C-reactive protein	32,000	0.3	5.2	6.9
No test	35,000	0.3	5.7	7.9

**Table 9 biomedicines-11-02156-t009:** Total LoS of 30 days for early and symptomatically treated patients.

	Total Cost [EUR (Rounded off)]	Deaths	Mean Days ICU	Mean Days General Ward
Calprotectin immunoassay	29,000	0.2	3.7	14.7
WBC count	36,000	0.3	4.9	15.1
Procalcitonin	37,000	0.3	5.1	15.3
C-reactive protein	37,000	0.3	5.2	14.9
No test	40,000	0.3	5.7	15.8

## Data Availability

This study is based on a previously published study [16] that was ethically approved.

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
