# Peer review of "Analysis of Calprotectin as an Early Marker of Infections Is Economically Advantageous in Intensive Care-Treated Patients"

_biomedicines, 2023, doi:10.3390/biomedicines11082156_

Round 1
Reviewer 1 Report
I enjoyed reading this very interesting paper on cost-effectiveness of calprotectin used as an early biomarker of infections. If I understand correctly, the paper is done on the premise that other markers would not be used synchronously in this context (thus users would save money by doing calprotectin instead of other investigated parameters - WBC, procalcitonin, CRP). As authors may be aware, it is very unlikely that ICU patients would not do CRP and WBC as a part of routine testing in the context of various diseases that require ICU admission. I suggest the authors to provide additional scenarios regarding cost-effectiveness in a way to compare calprotectin, WBC, procalcitonin and CRP as individual parameters with additional combinations of WBC + CRP, WBC + CRP + procalcitonin and calprotectin + WBC, calprotectin + WBC + CRP, calprotectin + WBC + CRP + procalcitonin.
Author Response
I enjoyed reading this very interesting paper on cost-effectiveness of calprotectin used as an early biomarker of infections.
If I understand correctly, the paper is done on the premise that other markers would not be used synchronously in this context (thus users would save money by doing calprotectin instead of other investigated parameters - WBC, procalcitonin, CRP).
A: Dear Reviewer! Thank you very much for kind and most valuable comments, which will help us to further improve this manuscript. Obviously, we have not been sufficiently clear regarding the use of calprotectin as a biomarker in ICU patients. We do not aim to abolish, or even to reduce, the use of other biomarkers used as biomarkers of infections.
Our intention was to evaluate whether calprotectin, used as a biomarker of infection in ICU patients, could serve as a complement to conventionally used parameters of infection.
Since there is a possibility for misunderstanding it is now mentioned in both Introduction and Discussion, that the intended use of calprotectin is complementary.
As authors may be aware, it is very unlikely that ICU patients would not do CRP and WBC as a part of routine testing in the context of various diseases that require ICU admission.
A: We fully agree with this point of view, since the focus of various biomarkers of infections are different. The advantage of calprotectin in this context is faster increase of the biomarker in circulation and earlier detection of infection, when compared to other routinely used biomarkers.
I suggest the authors to provide additional scenarios regarding cost-effectiveness in a way to compare calprotectin, WBC, procalcitonin and CRP as individual parameters with additional combinations of WBC + CRP, WBC + CRP + procalcitonin and calprotectin + WBC, calprotectin + WBC + CRP, calprotectin + WBC + CRP + procalcitonin.
A: This is a very interesting thought, but from a practical aspect (impact of such calculations on outcome) the material is too limited to allow us to make a reliable determination of these comparisons. A follow-up study on this issue would be a possible solution. Since it may be regarded that the lack of comparisons between calprotectin versus calprotectin in combination with the other biomarkers is a draw-back of our study, this is now mentioned in the Limitations section of the manuscript.
Reviewer 2 Report
In general, the authors did a good job on this research and the manuscript is written logically, with the overall claim supported by the results. The article's structure is simple, with a broad discussion of the importance of Calprotectin as an early determinant of bacterial infections, which may serve as a diagnostic tool facilitating decision-making on antibiotic treatment.
This study hypothesizes that if Calprotectin were used predictively for 50% of patients, the cost-effectiveness of the treatments could save between 6,000-7,000 euro per patient, based on reduced stay in the ICU and general ward, respectively. The general concept of this study is interesting. It opens an avenue towards doctors to use Calprotectin as a tool for early diagnostic of bacterial infections and for scientists to find new point-of-care methods for calprotectin detection.
I don't have concerns about the scientific part, and the results of the current study are interesting.
Thank you.
Author Response
Answer to Reviewer 2.
Dear Reviewer! Thank you very much for your kind and encouraging words, putting our findings in a broad and valuable perspective. Since economical perspectives on healthcare seem to be increasingly important, we postulate that evaluations on cost–benefit will be gradually more and more considered by healthcare providers. It may be hypothesized that algorithms aiming to facilitate diagnostic procedures and clinical decision-making, managed by artificial intelligence may be progressively more important. Hence, we hope that our findings, presented in this article, may be of value in this paradigm.
Round 2
Reviewer 1 Report
thank You for these responses. I find them acceptable